# Monitoring the reproductive number of COVID-19 in France: Comparative estimates from three datasets

Christophe Bonaldi[1]*, Anne Fouillet[1], Cécile Sommen[1], Daniel Lévy-Bruhl[2], Juliette Paireau[2,3]

**1** Data Science Division, Santé Publique France, The French Public Health Agency, Saint Maurice, France, **2** Infectious Diseases Division, Santé Publique France, The French Public Health Agency, Saint Maurice, France, **3** Mathematical Modelling of Infectious Diseases Unit, Institut Pasteur, Université de Paris Cité, CNRS UMR 2000, Paris, France

* Christophe.BONALDI@santepubliquefrance.fr

**Data Availability Statement:** The data underlying the results presented in the study are available from https://github.com/christophe-bonaldi/Monitoring-R-effective.

## Abstract

### Background

The effective reproduction number (R$t$) quantifies the average number of secondary cases caused by one person with an infectious disease. Near-real-time monitoring of R$t$ during an outbreak is a major indicator used to monitor changes in disease transmission and assess the effectiveness of interventions. The estimation of R$t$ usually requires the identification of infected cases in the population, which can prove challenging with the available data, especially when asymptomatic people or with mild symptoms are not usually screened. The purpose of this study was to perform sensitivity analysis of R$t$ estimates for COVID-19 surveillance in France based on three data sources with different sensitivities and specificities for identifying infected cases.

### Methods

We applied a statistical method developed by Cori et al. to estimate R$t$ using (1) confirmed cases identified from positive virological tests in the population, (2) suspected cases recorded by a national network of emergency departments, and (3) COVID-19 hospital admissions recorded by a national administrative system to manage hospital organization.

### Results

R$t$ estimates in France from May 27, 2020, to August 12, 2022, showed similar temporal trends regardless of the dataset. Estimates based on the daily number of confirmed cases provided an earlier signal than the two other sources, with an average lag of 3 and 6 days for estimates based on emergency department visits and hospital admissions, respectively.

### Conclusion

The COVID-19 experience confirmed that monitoring temporal changes in R$t$ was a key indicator to help the public health authorities control the outbreak in real time. However, gaining

**Funding:** The authors received no specific funding for this work.

**Competing interests:** The authors have declared that no competing interests exist.

access to data on all infected people in the population in order to estimate R$t$ is not straight-forward in practice. As this analysis has shown, the opportunity to use more readily available data to estimate R$t$ trends, provided that it is highly correlated with the spread of infection, provides a practical solution for monitoring the COVID-19 pandemic and indeed any other epidemic.

## Introduction

The effective (or time-varying) reproduction number, R$t$, represents the average number of secondary cases at time t of an outbreak caused by an infected person during the infectious period. If R$t$ is greater than 1, each infected person transmits the virus to more than one person on average, meaning that the epidemic grows. If R$t$ is less than 1, an infected person infects less than one person on average, resulting in the decline of the epidemic.

The near real-time estimation of R$t$ makes it possible to monitor the outbreak over time and evaluate the effectiveness of strategies implemented to control the infection by reducing R$t$ and maintaining it to values below 1 [1,2]. As the COVID-19 outbreak has shown, along with other available epidemiological data, R constitutes also an important indicator to help guide appropriate health policies and policymakers decisions [3–7].

Several methods have been developed to estimate R$t$ in an epidemic context, although the majority are based on transmission models [8–10]. These require many parameters, time, and expertise or incidence data from times later than *t* which limit their use in real practice. Accordingly, these methods do not allow for the real-time or near real-time monitoring of variations in R$t$. In 2013, a team of modelers from Imperial College London published a simpler method for estimating the effective reproduction number [11,12]. Drawing on this method, Santé publique France, the French Agency for Public Health, has produced daily R$t$ values at the national and regional levels to monitor the spread of COVID-19 in near real-time since May 27, 2020. The aim of this paper is to assess the ability of the method to monitor variations in R$t$ based on three data sources with different levels of sensitivity and specificity.

## Methods

### Data collection

To monitor the spread of COVID-19 in France, three data sources were used to estimate R$t$. The daily number of confirmed cases of COVID-19 was obtained from the SI-DEP database (*Système d'Information de Dépistage en Population*–Population-based Screening Information System), which records data on all virological testing carried out in France (metropolitan and overseas territories) to diagnose COVID-19 [13]. More precisely, SI-DEP includes the results of reverse transcriptase-polymerase chain reaction (RT-PCR) tests carried out by all medical laboratories and hospitals for the diagnosis of SARS-CoV-2, as well as antigenic tests performed by medical laboratories, pharmacists, physicians, and nurses. The SI-DEP system, which was specifically created to monitor the COVID-19 pandemic, has been fully operational since May 13, 2020. In France, the main indications for conducting COVID-19 tests were to confirm a diagnosis of suspected COVID-19, to carry out a control test to end isolation, or to obtain a COVID-19 certificate. In addition, local screening strategies could be implemented on an ad hoc basis, particularly in the case of clusters.

R$t$ was also estimated from the OSCOUR network (*Organisation de la surveillance coordonnée des urgences*–Coordinated emergency surveillance network). This national surveillance

network, which collects emergency department (ED) visits from about 700 French EDs (94% of national all-cause ED visits), is the main component of syndrome-based surveillance of ED carried out by Santé publique France [14,15]. We used the daily number of ED visits with suspected COVID-19 diagnosed by the emergency physicians taking care of the patient. To standardize practices for the classification code of suspected COVID-19 in the system and to ensure consistency in the diagnosis of COVID-19 by physicians, some recommendations detailing the clinical picture of patients suspected of being infected were sent to all EDs. These recommendations were important at the start of the pandemic, as COVID-19 was an emerging disease. The symptoms were not specific and could be confused with other illnesses such as influenza.

Finally, the third data source was the daily number of hospital admissions for COVID-19 recorded in the SI-VIC database (*Système d'information unique des victimes*–National victim information system). The SI-VIC system is primarily used to help health authorities monitor the number of persons hospitalized because of a health emergency and anticipate its consequences for the organization of hospitals. We included admissions to both general wards and intensive care units but excluded emergency, psychiatric, long-term, and rehabilitation care.

These three datasets differ in terms of their sensitivity and specificity to identify COVID-19 cases in the population. Not all infections are tested and thus included in the SI-DEP (sensitivity default). The OSCOUR network includes symptomatic individuals, often not tested in the ED, with a proportion not being COVID cases (specificity default). The COVID-19 patients seen in the ED are also likely to have more severe symptoms such as respiratory distress as opposed to moderate or mild symptoms (sensibility default). SI-VIC data were highly specific– all hospitalized patients were confirmed COVID-19 cases–but it only recorded hospitalized COVID-19 cases.

## Statistical method

The method used for estimating the time-varying reproduction number based on Cori et al. [11] was implemented in R software (version 4.1.1, R Foundation for Statistical Computing, Vienna, Austria) using the *EpiEstim* package.

*EpiEstim* is considered the most computationally efficient and ready-to-use software that is currently available [16,17]. In short, the method can be divided into two main steps. First, the time series of daily counts of patients were smoothed using cubic smoothing splines to remove random daily variations and weekend effects. Second, the reproduction number was estimated with *EpiEstim* using a 7-day sliding window and under the assumption that the serial interval (time between symptom onset in an index case and symptom onset in a secondary case) followed a gamma distribution with a mean of 7 days and a standard deviation of 5.2 days. The parameters for the gamma distribution were fixed according to the model by Salje et al. [18]. Uncertainty intervals for R$t$ were provided by the 95% credibility intervals from the estimated R$t$ posterior distribution in the Bayesian framework of the *EpiEstim* method. A comparison of the daily patterns of the estimated R$t$ from the three datasets ran from May 27, 2020, to August 12, 2022.

## Results

Fig 1 shows the daily case counts recorded in each database in metropolitan France from May 13, 2020, to August 12, 2022. As expected, the absolute number of cases reported by each system differed substantially, with much larger volumes for the SI-DEP data compared to the two other data sources. The three databases had a weekly cycle, although this was particularly evident for the SI-DEP data for which screening activity was down on weekends due to the weekly

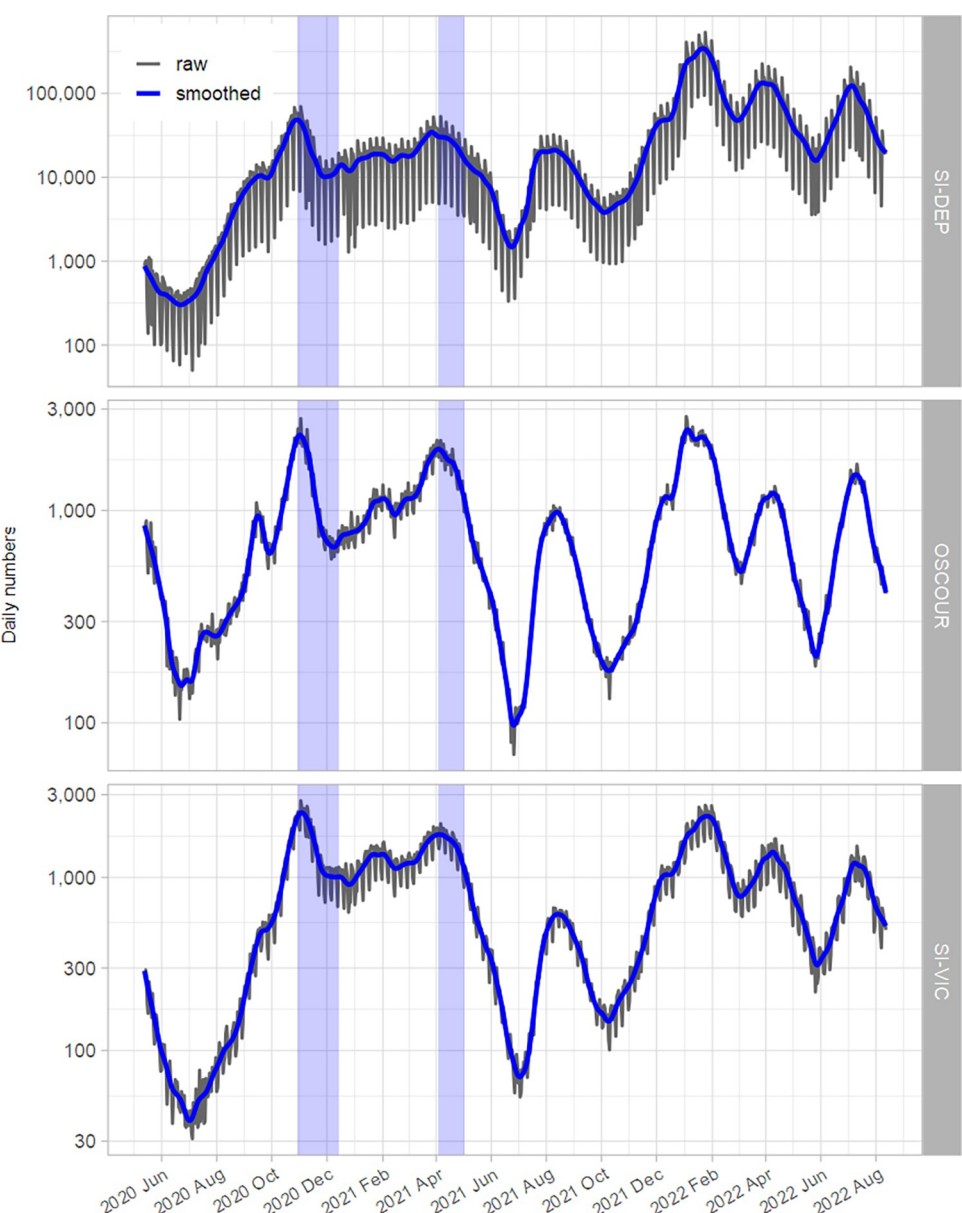

**Fig 1. Daily RT-PCR or antigenic confirmed COVID-19 cases (top), daily number of emergency department consultations with suspected COVID-19 (middle), and daily number of new hospital admissions for COVID-19 (bottom) between May 13, 2020, and August 12, 2022, in metropolitan France.** A base-10 log scale is used for the Y-axis. Blue curves are smoothed counts using cubic smoothing splines. Shaded areas indicate the second nationwide lockdown from October 29 to December 14, 2020, and the third from April 3rd to May 2nd, 2021.

closure of many private medical laboratories and pharmacists. Regardless of the level of the counts, temporal curves showed similar patterns, particularly when comparing OSCOUR and SI-VIC data. In the three data sources, a rapid decline in case counts occurred just after the implementation of national lockdowns to stop the second and third pandemic waves observed in France. The lockdown from October 29 to December 14, 2020 imposed confinement measures like movement restrictions, closure of non-essential shops but schools remained open. The lockdown between April 3rd and May 2nd, 2021 was less restrictive for movements and shopping but imposed closure of schools.

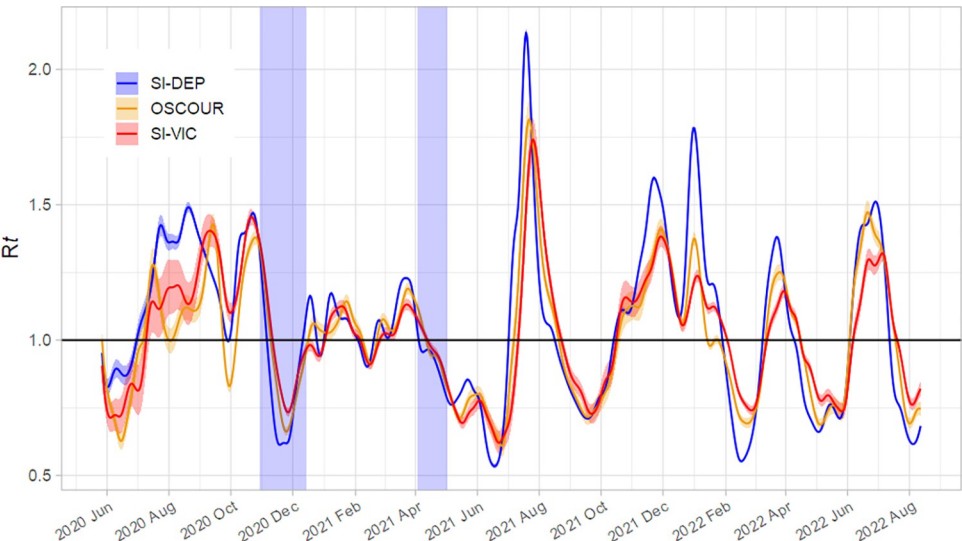

**Fig 2. Estimates of the daily reproduction number (Rt) throughout the COVID-19 pandemic from May 27, 2020, to August 12, 2022 in metropolitan France (mean as solid lines and 95% credible interval as shaded areas) based on confirmed cases among the tested population (SI-DEP), suspected cases in emergency department visits (OSCOUR), and hospitalized patients (SI-VIC).** Shaded areas indicate the nationwide lockdown periods (October 29 to December 14, 2020, and April 3rd to May 2nd, 2021).

Fig 2 shows the temporal variations of R*t* estimated from each dataset. As desired, R*t* decreased and fell below 1 after the implementation of nationwide lockdowns. However, during the second confinement, we observe a rebound in the estimate, which increases to a value close to 1 at the end of the confinement. For the entire period, the three curves were closed, except for a few weeks in 2020 between mid-July and late September. However, this period was marked by a sluggish increase in the pandemic, with a very small number of daily cases reported by the three information systems (Fig 1). In particular, this resulted in wider credibility intervals for the R*t* estimates.

Fig 3 shows the distribution of the differences between the daily estimates calculated using the three datasets. The median differences were similar and close to zero, with the extreme differences observed near the peaks and troughs of the time series for the R*t* values (Fig 2). Overall, a slight time lag is perceptible between the R*t* curve estimated from SI-DEP and the curves estimated from the two other datasets: based on the maximum correlation, a lag of 3 and 6 days was observed for OSCOUR and SI-VIC, respectively (Table 1).

## Discussion

In France, the estimates of the instantaneous reproduction number R*t* used to monitor the spread of COVID-19 showed similar trends regardless of the dataset used: confirmed cases among the tested population (SI-DEP), suspected cases in ED visits (OSCOUR), or hospitalized patients (SI-VIC). Estimates based on the daily number of cases identified from the virological test results (SI-DEP) provided an earlier signal that the two other data sources. This finding was predictable due to the nature of the data recorded by the three information systems, which collect data on cases at different infection stages. Indeed, the average time between symptom onset and testing is around 1–2 days (mean delay observed in the SI-DEP data), while hospitalizations occur approximately 6 to 7 days after the onset of symptoms [18]. This is consistent with the lag of 6 days observed between R*t* estimated from hospitalization data (SI-VIC) and R*t* estimated from virological test results (SI-DEP). Cases collected during an ED

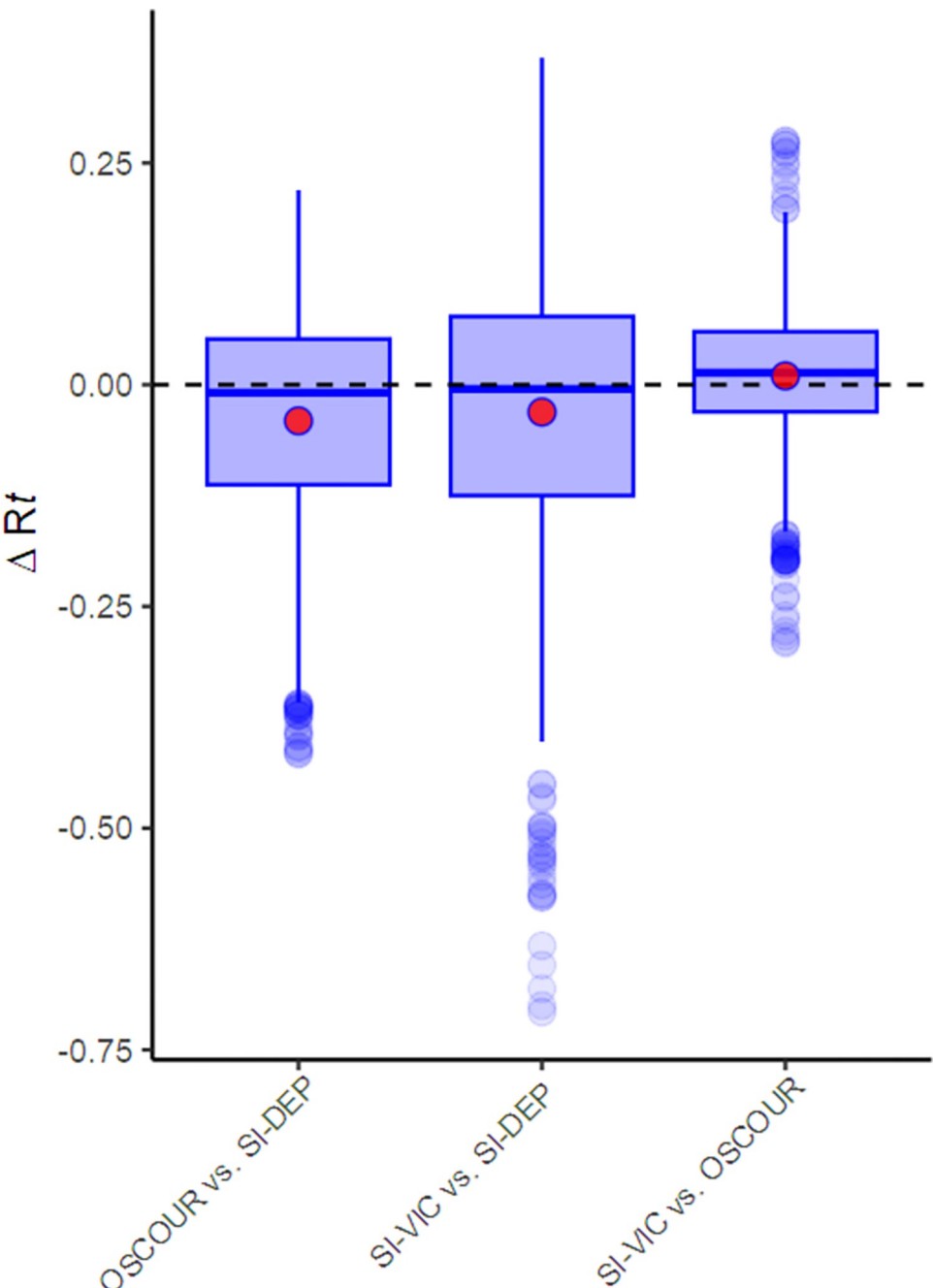

**Fig 3. Boxplot of the differences between the daily estimates of Rt depending on the datasets used from May 26, 2020, to August 12, 2022, in metropolitan France.** Red points highlight the mean differences.

visit generally corresponded to patients with deteriorating symptoms such as respiratory distress, which often take place after symptom onset but before possible hospitalization, which is consistent with the slight lag of 3 days in R$t$ estimated with OSCOUR compared to SI-DEP estimates.

As noted, the major discrepancies observed for a few weeks in 2020 (from mid-July to September) between the three estimates corresponded to a period of low incidence. During this period, an accurate estimate was not possible, because the R$t$ values were particularly volatile.

**Table 1. Pearson correlation between the daily reproduction numbers (R*t*) for COVID-19 from May 27, 2020, to August 12, 2022, in metropolitan France, estimated based on confirmed positive cases in the tested population (SI-DEP), suspected cases in emergency department visits (OSCOUR), and hospitalized patients (SI-VIC) according to the temporal lag (in days).** The bold figures highlight the maximum correlation.

| SI-DEP | OSCOUR | | Lag (days) | | | | | |
|---|---|---|---|---|---|---|---|---|
| | | 0 | 1 | 2 | 3 | 4 | 5 |
| | Correlation | .889 | .906 | .916 | **.920** | .918 | .910 |
| | SI-VIC | | Lag (days) | | | | | |
| | | 0 | 4 | 5 | 6 | 7 | 8 |
| | Correlation | .822 | .907 | .916 | **.921** | .920 | .914 |

Various factors such as active case finding and contact tracing during this period with the identification of outbreak clusters (mainly asymptomatic people) can cause large variations in the number of confirmed cases. In addition, for a few months at the start of the pandemic until the end of 2020, a significant number of cases reported by the ED were not tested, although with the increasing availability of tests and especially antigen tests over time, an increasing proportion of cases notified through the OSCOUR system were confirmed cases. This might explain why the R*t* series became more similar over time.

The instantaneous reproduction number found using the results of virological tests (SI-DEP) provided higher peak and trough values than the estimates based on the number of suspected cases in ED visits or hospitalized patients. The number of confirmed cases in tested patients are sensitive to factors such as screening control strategies or contact behaviors of confirmed cases, which could explain why the peak and trough values are higher. Differences between the extreme values of the R*t* estimates seem notably higher since late 2021. This period corresponds to the emergence of three factors influencing the evolution of the pandemic: (1) the greater use of spontaneous testing before and during the Christmas and New Year period, which increased the number of identified positive cases, particularly asymptomatic cases; (2) the effect of vaccinating a significant proportion of the population, especially the most vulnerable; and (3) the emergence of more contagious but less severe variants, thereby reducing the incidence of severe cases in ED and hospitals. As a result, the R*t* estimates showed greater differences in the extreme values, although the overall trend remained robust regardless of the data used for the estimation.

Over the entire period studied, estimates of R*t* have fluctuated constantly. The largest waves are correlated with the emergence of new variants such as the Alpha variant at the end of 2020, the Delta variant in the summer of 2021 and the Omicron variant at the beginning of 2022. Contextual events such as the Christmas and New Year period could explain the peak observed in 2021. We also noticed that during the second lockdown, the number of reproductions had surprisingly rebounded after reaching a minimum value, returning to a value close to 1 when the lockdown ended. The second national lockdown was less strict than the first one (from March 17 to May 3, 2020), but all schools remained open. This could suggest that children and adolescents—who present a low risk of infection or if they are infected it is likely to be mild—may have been transmitters of the virus into the household during this period. It should be noted that we did not observe this phenomenon during the third lockdown, which was characterized by a reversed strategy with the early closure of schools before the Easter school holidays.

The *EpiEstim* package provided a straightforward method to estimate time-varying instantaneous reproduction numbers using time series of incident cases (confirmed cases, ED visits, or hospitalizations). We retrospectively showed that using this method to monitor the dynamics of the COVID-19 pandemic in France was effective in detecting temporal variations in the

reproduction number and monitoring the effectiveness of interventions. The implementation of *EpiEstim* requires knowledge of the serial interval distribution, which we assumed to be gamma distributed with a mean of 7 days and a standard deviation of 5.2 days based on the model by Salje et al [18] and consistent with the analysis of the first transmission chains in France [19]. We used the same distribution over the whole study period. One limitation of this approach is that it does not account for temporal variations in the serial interval distribution. Such variations can occur due to changes in people's behavior or policy interventions adopted in a country to limit the spread of the disease [20]. For instance, it has been demonstrated that case isolation can reduce the serial interval [21]. Other studies found lower serial intervals such as around 5 days in the meta-analysis of Rai et al [22]. In addition, recent studies reported even shorter serial intervals in SARS-CoV-2 with the new Omicron variant—appeared in France at the end of November 2021—compared to the Delta variant [23,24]. Varying the distribution parameters of the serial interval can affect the magnitude of the daily estimates of R$t$ with higher values (when R$t > 1$) when the mean increased [20]. Conversely, as the standard deviation of the serial interval increases, the estimate of Rt decreases [25] (see Supporting information). It should be noted that in Cori's method, the serial interval is used as a substitute for the generation interval (the time elapsed between the infections of an infector and an infected person). But replacing the generation interval with the serial interval tends to bias the estimate of Rt [26]. In particular, when the variance of the serial interval is higher than that of the generation interval, Rt will consequently be underestimated. As illustrated in the article by Ganyani et al. [27] for COVID-19, the standard deviation of the generation interval is generally smaller than that of the serial interval. However, this does not affect the time when R$t$ crosses the threshold of 1, which is one of the most important indicators for public health decision-makers.

Over the whole period, the estimated R$t$ does not exceed the value of 2.1, obtained from SI-DEP data in July 2021. The new Omicron variant that appeared in France at the end of November 2021 was supposed to be much more contagious than the Delta variant, which was the majority variant. However, when this new variant appeared, the peaks in the R$t$ did not exceed 1.8, at the end of December 2021. This observation is not consistent with the average R$t$ of 3.4 reported by Liu et al. [26] for the Omicron variant. However, we can assume that the impact of vaccination—over 75% of the French population had full vaccination coverage by December 2021—limited the spread of the virus. The estimated Rt was therefore much lower than the peak observed in July 2021, when the delta variant was in the majority and vaccination coverage was lower. In addition, there were 2 major waves: the first starting in November, but probably still linked to the delta variant, and a rebound in mid-December linked to the spread of the Omicron variant in combination with the end-of-year festivities. Again, immunity induced by the vaccine or contamination probably helped to limit the spread of the variant. Finally, it should also be noted that preventive measures like masks on public transport, widespread teleworking, reception of a limited number of customers depending on the surface area in shops or subject to vaccination certificate, etc. were still in force at that time in France (until 12 February 2021 for the most part). This may also have helped to limit the spread of this variant in France.

The size of the sliding window over which the R$t$ estimates are calculated was fixed to 7 days in our analysis. This time window acts as a smoothing parameter with more pronounced smoothing (along with narrower credible intervals) as the window increases in size [11]. However, a wider window makes the estimates less responsive which means it constrains the ability to observe the most recent changes in the reproduction numbers. In our case, a 7-day sliding window was considered a good compromise between responsiveness and smoothness of the estimates.

The estimation of the reproduction number is theoretically based on all infected cases in the population. In practice, however, it is often difficult to identify all infected cases in the population [16], particularly when a large proportion does not show symptoms as in the COVID-19 outbreak. Cori et al. [11] demonstrated with simulations that their method is robust as long as the proportion of detected cases or the fraction of asymptomatic cases remains stable over time, which we assumed in our study. In addition, this analysis shows that under the current conditions with a long observation period, the method can be reasonably applied to data such as hospitalized patients or ED reports. We can therefore assume that all the data strongly correlated with the number of infected cases in the population are suitable for estimating the $Rt$ temporal trends, provided that this correlation remains relatively stable over time.

Monitoring changes in the $Rt$ has proven to be a major indicator for guiding public health authorities and stakeholders for fitting infection control strategies in real time. In France, variations in the $Rt$ were widely and regularly communicated to the general public, with the $Rt$ becoming a simple pedagogical tool to explain the need to reinforce individual preventive behaviors or implement collective protective measures when the number of reproductions exceeded 1. The use of the popular package *EpiEstim* to estimate the daily $Rt$ with input data of varying quality since the beginning of the outbreak showed the robustness of the method. Daily data relating to the total number of infected cases in a population is usually not available during a large outbreak. This may require the development of infrastructure to test people and record the test results, which is often challenging to set up, particularly in some developing countries. Along with the simplicity of the method of Cori et al., the opportunity to use easy-to-obtain data from ED consultations or hospitalizations, provided that these data are strongly correlated with the propagation of the infection, provides a concrete solution for monitoring the COVID-19 pandemic and, more broadly, any epidemic.

## Supporting information

**S1 File. Details on the Cori's method for estimating effective reproduction numbers.** (DOCX)

## Author Contributions

**Conceptualization:** Christophe Bonaldi, Anne Fouillet, Juliette Paireau.

**Data curation:** Christophe Bonaldi, Anne Fouillet, Cécile Sommen, Juliette Paireau.

**Formal analysis:** Christophe Bonaldi, Anne Fouillet, Daniel Lévy-Bruhl, Juliette Paireau.

**Methodology:** Christophe Bonaldi, Juliette Paireau.

**Supervision:** Daniel Lévy-Bruhl.

**Validation:** Cécile Sommen.

**Writing – original draft:** Christophe Bonaldi.

**Writing – review & editing:** Christophe Bonaldi, Anne Fouillet, Cécile Sommen, Daniel Lévy-Bruhl, Juliette Paireau.

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
