## [Decision Letter · Decision Letter 0]

11 Jul 2023

PONE-D-22-25321Monitoring the reproductive number of COVID-19 in France: Comparative estimates from three datasetsPLOS ONE

Dear Dr. Bonaldi,

Thank you for submitting your manuscript to PLOS ONE. After careful consideration, we feel that it has merit but does not fully meet PLOS ONE’s publication criteria as it currently stands. Therefore, we invite you to submit a revised version of the manuscript that addresses the points raised during the review process.

Please excuse that this review took so long. I as an editor took over this paper late/recently, and then it was really difficult to find reviewers on my part as well. Please revise the manuscript according to all points of both reviewers. Especially the points of Reviewer 1 should be addressed as well as possible, since his/her recommendation was "reject". Thank you!

We look forward to receiving your revised manuscript.

Kind regards,

Kaspar Staub

Academic Editor

PLOS ONE

Journal Requirements:

Additional Editor Comments (if provided):

As stated above: Please excuse that this review took so long. I as an editor took over this paper late/recently, and then it was really difficult to find reviewers on my part as well. Please revise the manuscript according to all points of both reviewers. Especially the points of Reviewer 1 should be addressed as well as possible, since his/her recommendation was "reject". Thank you!

Reviewers' comments:

Reviewer's Responses to Questions

**Comments to the Author**

1. Is the manuscript technically sound, and do the data support the conclusions?

Reviewer #1: No

Reviewer #2: Yes

2. Has the statistical analysis been performed appropriately and rigorously? 

Reviewer #1: I Don't Know

Reviewer #2: Yes

3. Have the authors made all data underlying the findings in their manuscript fully available?

Reviewer #1: Yes

Reviewer #2: Yes

4. Is the manuscript presented in an intelligible fashion and written in standard English?

Reviewer #1: No

Reviewer #2: Yes

5. Review Comments to the Author

Reviewer #1: Thank you for the opportunity to review this manuscript.

The authors apply a commonly-used method to three different input sources. The authors seem to be using a “throwing data sets at the wall and see what sticks”-approach. I wonder whether this would be better served as a report from the public health agency rather than a peer-reviewed scientific manuscript; as there does not seem to be much substance for researchers outside of that agency.

Additional comments:

I would suggest the authors use the short title “Monitoring the reproduction number of COVID-19 in France” rather than “Monitoring the reproduction number” as it makes more sense to split at the colon

The authors should provide meta data with their data (following a meta data schema) and should consider depositing their data in a repository such as Zenodo such that it can have a persistent identifier (which can then be used to reference where the data can be accessed). As the authors are using GitHub, there should be an easy plug in allowing for this without too much effort.

Line 25: The authors should explain what they mean by “near-real-time” as this is not a common term nor evident from their manuscript. “Real-time” would imply evaluating the next time point (immediately following) but it is unclear what “near-real-time” would be.

Line 28: The authors should describe why it “proves challenging”. Presumably due to asymptomatic cases but they could be explicit about it.

Line 54: The authors could explain that the effective reproduction number R_t is not the only indicator policy makers might be interested in as it is not informative by itself; additional context is required. For example: an R_t > 1 could be problematic but if additional surge capacity is available in hospitals or most cases not severe does not have to be an issue.

Line 57: Is it true that transmission models do not allow for real-time estimation of R_t? It seems to be that there may be contemporary COVID-19-related literature which is relevant here. The references provided by the authors are from the 2000s and more work may have been done in 2020-2022.

Line 61-62: “The beginning of the outbreak” is not specific; please provide the date.

Line 74: Is the SI-DEP system expected to undergo long-term maintenance or will this data become obsolete in future? Authors need to ensure reproducibility of their work

Line 84-86: It is unclear what this means

Line 91-92: Were “emergency” units excluded to avoid overlap with the casualty department data?

Line 93: Provide estimates of the sensitivity and specificity rates please. If the authors wanted to provide a scientific investigation/modelling study rather than a data analysis they could examine the effects of a higher/lower value of these rates. If doing so a study protocol outlining how the data for examining such a question would be simulated should be deposited to avoid poor scientific practices. The PLOS family of journals supports registered reports, which might be an option to consider.

Line 106: Where is the data from these data bases made available (to ensure reproducibility)?

Line 112-113: Please provide version numbers and settings used (seeds for pseudo-random number generators etc.)

Line 114: A citation is needed for this as well as an explanation of which metrics this statement is made according to

Line 116-117: Why not use a weekly aggregated time series in place of this? If “near-real-time” estimation is required, this would presumably be alright (and remove the weekly oscillations in case counts)

Line 120: Provide formulae

Line 121: Authors should explain why they chose a 95% level

Line 130-139: This is not a result but rather a description of the data inputs. Accordingly, it should be moved to the methods section.

Line 135-136: This is moreso true for the OSCOUR and SI-VIC data sets which could be mentioned

Line 137: The format of the lockdowns should be explained as this is country-specific and the reader may not know what restrictions were in place in France

Line 137-139: As this is outside of the study period, it should be removed

Line 146-147: See previous comment

Line 153-154: But the intervals seem to be non-existent the rest of the time. Is that plausible?

Line 162: When this has a temporal dimension, why not show it?

Line 169-171: Provide better axis labels

Line 172: Which type of correlation is used. Provide the formulae

Line 182: But how confident can we be that the signal is correct?

Line 182-191: This intuition could be laid out in the beginning

Line 197: When is “a few months at the start of the pandemic”? Please provide dates

Line 216-218: But how did you quantify this?

Line 227-228: Please provide dates for the emergency for France since this is context-specific (depends on local conditions)

Line 228-230: Why not do a sensitivity analysis? Change point analysis? Interrupted time series analysis?

Line 235: What does “less responsive” mean?

Line 242-243: But you noted in line 207-209 that there was a change in the ascertainment (which is super interesting) as your reasoning so that does not seem consistent with this assumption

Line 246-247: Would this be included in a Sante Publique France early warning system?

Line 261: What about waste water monitoring? Flunet? Other non-invasive options for estimating population-level incidence and prevalence exist

Provide an author CReDIT (contributor roles taxonomy) statement

Formatting:

The authors should ensure consistency of $R_t$ throughout (R$t$ and R) seen as well as the naming (“instantaneous reproduction number” also used in place of “effective reproduction number”). Additionally, the use of R in the introduction is inconsistent with the abstract

Line 60: I would suggest the authors use the name Sante Publique France throughout (since it is introduced)

Line 77: I would suggest using the term “COVID-19 certificate” rather than “health pass” as the former is more known (being the term used at EU-level)

Line 81-82: It should be “EDs” since multiple departments are being referenced

Line 84: I would suggest authors used “triage” in place of “coding” to avoid confusion with the concept of coding as it pertains to software code

Line 90: I would suggest “health emergency” in place of “unusual health situation”

Line 141: This should be “RT-PCR” to be consistent with when the abbreviation was introduced

Line 149: R_t falling below 1 is “desired” rather than “expected”

Line 156: Should be “Fig 2”

Line 272: Should be “Imperial College”

In abstract should be date accessed rather than date cited

Line 299 and 305: What is “80-“ is there a superfluous hyphen?

Check colours are differentiable when printing in black and white

In figure 3 could include a dashed line at zero since this is the value of interest for this graph (though perhaps a table would be better)

Reviewer #2: This paper applies a method developed by Cori et al (2013) and implemented in the statistical software R to calculate the effective reproduction number Rt during the COVID pandemic in France (May 2020-March 2022) on a daily basis. This method requires some general information on the distribution of the serial interval for the COVID-19, as well as daily incidence data of COVID-19. For the latter, three distinct data sets were used, respectively the daily number of 1) confirmed cases (among those tested), 2) emergency department visits (with suspected COVID), and 3) hospital admissions (due to COVID). Although such data are necessarily incomplete, the Rt obtained were remarkably similar using any of the three data sets, illustrating the fact that a real-time surveillance can be carried out with imperfect data, a useful result for public health.

The paper is very well written and clearly publishable. A minor revision would make it even better. I have the following remarks:

1) Statistical method: If feasible, a little more explanation about the method of Cori et al (possibly citing the related paper by Wallinga and Teunis, 2004, American Journal of Epidemiology, 160, 509-516) would be welcome for an interested reader.

2) Discussion: It is written that Rt becomes higher as the mean and standard deviation of the serial interval increase. While this is true for the mean (at least when Rt is greater than 1), Rt would actually decrease with a higher standard deviation, being e.g. overestimated in the extreme case where the standard deviation is wrongly set to 0 (see e.g. Wallinga & Lipsitsch, 2007, Proc. of the Royal Society B, 274, 599-604). This should be corrected.

3) As acknowledged by Cori et al (2013), the serial interval (time between onset of symptoms in the infector and the infected) is used as a proxy of the generation interval (time between infections of the infector and the infected) which should be ideally used in their method. As a direct consequence of my first comment, this should result in an underestimation of Rt since the standard deviation of the generation interval (which should be used) is generally smaller than that of the serial interval (which has been used, see also Ganyani et al, 2020, Euro Surveillance, 25, 2000257). This should be mentioned in the Discussion.

4) In the Discussion, it is briefly mentioned that serial intervals could be shorter with the new Omicron variant. But a shorter serial interval would lead to a smaller Rt, although the Omicron variant was supposed to be much more contagious than the original one, reaching an average Rt of 3.4 (Liu and Rocklov, J. Travel Medicine, 2022, 29, 1-4). How do the authors explain that the peaks in their estimation of the reproduction number did not get that high when new (more contagious) variants appeared? This would deserve one (possibly short) paragraph in the Discussion.

5) Results: “As expected, Rt fell below 1 after the implementation of nationwide lockdown”. It is in fact not obvious to see the effects of the lockdown on such a graphic. These were not the only occasions where Rt fell below 1. Note also that the reproduction number started to rise again before the end of the first lockdown, while it continued to fall well after the second lockdown. One or two sentences recognizing this could be an option (public health is a challenging science!).

6) Caption Figure 2: Write Fig.2 instead of Fig.1. Would you also like to add the relevant dates here (May 2020-March 2022)?

6. PLOS authors have the option to publish the peer review history of their article (what does this mean?). If published, this will include your full peer review and any attached files.

Reviewer #1: No

Reviewer #2: No

---

## [Author Response · Author response to Decision Letter 0]

8 Sep 2023

Dear Dr Staub, 

We would like to thank you for giving us the opportunity to revise our manuscript despite the negative recommendation of Reviewer 1. Reviewer 2 disagreed and suggested improvements to clarify the manuscript. Both reviewers provided useful comments, which we have tried to take into account in order to improve the quality of our manuscript. We have therefore responded to the reviewers' comments and modified the manuscript accordingly. In particular, we hope that our responses will convince reviewer 1 to change his mind about our work.

As desired, there are two manuscript files; one with track changes (in yellow font) and one clean version.

Yours sincerely,

Christophe Bonaldi, on behalf of co-authors

Reviewers' comments:

Reviewer #1: Thank you for the opportunity to review this manuscript.

The authors apply a commonly-used method to three different input sources. The authors seem to be using a “throwing data sets at the wall and see what sticks”-approach. I wonder whether this would be better served as a report from the public health agency rather than a peer-reviewed scientific manuscript; as there does not seem to be much substance for researchers outside of that agency.

Authors’ answer: The Reviewer questions the scientific value of this work, reducing it to a simple question of interest for a public health agency. We are obviously rather surprised by this particularly harsh opinion of our work. Our goal was to show that a real-time surveillance can be carried out with imperfect data, collected in real conditions. Indeed, with our experience over a long period of the Covid-19 pandemic, we observed that the indicator of interest in this manuscript (the Rt) could be remarkably closed, even when using incomplete and imperfect data but easy to obtain. We still think that these findings are useful for not only public health agency but also researchers or modellers who use this indicator, in example for prediction models. Furthermore, although this paper focuses specifically on the Covid-19 pandemic, the proposed approach is fully reproducible and generalizable to the surveillance of any epidemic in a population. 

In the following, we have responded point by point to the Reviewer's comments; we hope that these responses will clear up any ambiguities and lead the Reviewer to modify his point of view on this work.

Additional comments:

I would suggest the authors use the short title “Monitoring the reproduction number of COVID-19 in France” rather than “Monitoring the reproduction number” as it makes more sense to split at the colon

Authors’ answer: The short title has been modified accordingly. 

The authors should provide meta data with their data (following a meta data schema) and should consider depositing their data in a repository such as Zenodo such that it can have a persistent identifier (which can then be used to reference where the data can be accessed). As the authors are using GitHub, there should be an easy plug in allowing for this without too much effort.

Authors’ answer: Thank you for this suggestion. We do not generally use this platform, but indeed, it has the advantage of providing a Digital Object Identifier (DOI) for the deposited data and code, so that they can be easily cited. We can simply plug this material from GitHub. 

Line 25: The authors should explain what they mean by “near-real-time” as this is not a common term nor evident from their manuscript. “Real-time” would imply evaluating the next time point (immediately following) but it is unclear what “near-real-time” would be.

Authors’ answer: Cori's method aims to estimate instantaneous numbers of reproduction varying over time, which makes it possible to measure transmission at a precise moment: today, for example, an estimate in "real-time". As in other publications (see the very interesting paper of Gostic et al., 2020 for example), we have preferred to use the term "near-real time" because it is closer to reality in practice. In fact, we rarely have the data to produce an estimate in real time, but data with a slight time lag - 2 or 3 days - to obtain an exhaustive count of cases (time required for data entry, transmission, processing in the database, etc.). We therefore prefer to keep this name, which in practice is the most honest. 

Line 28: The authors should describe why it “proves challenging”. Presumably due to asymptomatic cases but they could be explicit about it.

Authors’ answer: Yes in part, because the cases identified are only those confirmed by biological tests, but people who are asymptomatic or have mild, non-specific symptoms are not usually screened. But, above all, beyond the exhaustive identification of all infected cases in a population at national level, the collection and recording of the information in a database requires expensive systems, the collaboration of thousands of laboratories and means to enter, to transmit and to process the data in the database. The authors can testify that setting up a system such as SI-DEP and maintaining it on a daily basis during the COVID-19 pandemic was indeed a challenge…

The abstract has been modified as suggested by the Reviewer.

Line 54: The authors could explain that the effective reproduction number R_t is not the only indicator policy makers might be interested in as it is not informative by itself; additional context is required. For example: an R_t > 1 could be problematic but if additional surge capacity is available in hospitals or most cases not severe does not have to be an issue.

Authors’ answer: We fully agree with this comment. The effective reproduction number is of course not the only indicators and it cannot be interpreted on its own, but in relation to other available epidemiological data. As recommended, we add these supplementary elements in the revised manuscript.

Line 57: Is it true that transmission models do not allow for real-time estimation of R_t? It seems to be that there may be contemporary COVID-19-related literature which is relevant here. The references provided by the authors are from the 2000s and more work may have been done in 2020-2022.

Authors’ answer: Briefly, methods based on transmission models needs further assumption or data after time t (assumptions about future transmissibility), less suited to near real-time estimation of the reproduction number. Cori’s method uses only data before time t (new observed cases at successive times), which allow to compute an instantaneous reproductive number. To our knowledge (see for an exhaustive review Nash ID et al., 2022), no major methodological work was recently published to give an alternative to Cori’s method which is the reference method used in most countries during the Covid-19 pandemic. In addition, the paper of Gostic et al. published in Plos Computational Biology in 2020 studied the performance of different methods for estimating an instantaneous number of reproductions. Authors have concluded that Cori’s method stays the best method to estimate a (near)-real time Rt in practice. Nonetheless, as suggested by the Reviewer, we have completed the references in the revised manuscript. 

Line 61-62: “The beginning of the outbreak” is not specific; please provide the date.

Authors’ answer: We add the date as suggested by the Reviewer. 

Line 74: Is the SI-DEP system expected to undergo long-term maintenance or will this data become obsolete in future? Authors need to ensure reproducibility of their work

Authors’ answer: Although SI-DEP is still operational today, this system - which is specific to COVID-19 surveillance - is obviously not intended to last in the long term. Especially, medical and personal information collected in the system benefits - for ethical reasons - from some specific authorisations by French law in the framework of a health crisis response. They will lapse fairly as the epidemic situation improves. In addition, it is a costly and time-consuming system, requiring collaboration of thousands of laboratories (hospital, public, private) to feed data on a daily basis and at a national level. Furthermore, since September 2022, the data collected by this system has become increasingly unreliable due to a number of factors: the end of test reimbursement, altered transmissions from some laboratories, more frequent use of self-tests whose results escape the system, etc. Hence, the important issue raised in this article of being able to use a source of data that guarantees a certain stability over time.

Nonetheless, in this paper our aim was precisely to show that this type of data (results of biological test) is not essential for estimating instantaneous reproduction numbers but a real-time surveillance can be carried out with imperfect (but correlated) data, but easy to obtain. The OSCOUR network is for example a sustainable system. We focuses specifically on the Covid-19 pandemic that allow observing the robustness of estimates on a long period, but conclusions are likely generalizable to the surveillance of any epidemic in a population. For example, since the last winter season, Santé publique France has introducing the reproduction number as an indicator for the spread of influenza using Oscour data with some minor adaptation (parametrization) of the code provided in the Github repository. The same work will be carried out for bronchiolitis in the near future. Therefore, we think indeed this work is fully reproducible! 

Line 84-86: It is unclear what this means

Authors’ answer: To ensure consistency in the diagnosis of COVID-19 by physicians in emergency departments, recommendations detailing the clinical picture of patients suspected of being infected were sent to all emergency departments of the network. These recommendations were important at the start of the pandemic, as COVID-19 was an emerging disease. The symptoms were not specific and could be confused with other illnesses such as influenza. The sentence has been rewritten to make it less ambiguous. 

Line 91-92: Were “emergency” units excluded to avoid overlap with the casualty department data?

Authors’ answer: Indeed, some patients admitted to emergency departments were recorded in both the Oscour and SI-VIC systems. In particular, some patients were admitted to emergency before being hospitalised or admitted to intensive care. In this case, only the event corresponding to their admission to the general ward or intensive care unit was recorded. Other patients were admitted directly to hospital without passing through ED. It should be noted that the primary objective of Santé publique France's use of SI-VIC data was to identify the number of patients hospitalised for COVID-19, and it was this daily count that we used to estimate the Rt.

Line 93: Provide estimates of the sensitivity and specificity rates please. If the authors wanted to provide a scientific investigation/modelling study rather than a data analysis they could examine the effects of a higher/lower value of these rates. If doing so a study protocol outlining how the data for examining such a question would be simulated should be deposited to avoid poor scientific practices. The PLOS family of journals supports registered reports, which might be an option to consider.

Authors’ answer: Quantifying the sensitivity and specificity rates is impossible because we were not able to know all cases infected in the population. As highlighted before, an important part of cases was asymptomatic or has only mild symptoms. A meta-analysis published in JAMA (Ma et al., 2021) finds than less than 1% of these people were screened and globally, asymptomatic cases represent up to 40% of all cases. Of course, these figures vary greatly from country to country: depending on screening capacity, screening strategy, measures taken to curb the spread of the disease, etc… In addition, in France, a study suggested that a large number of symptomatic cases of COVID-19 were not screened too, despite the recommendations (Pullano et al., 2020). Nonetheless, the suggestion of the Reviewer is very interesting: based on simulated data, we could analysed the effect of sensibility and specificity on the reproduction number estimates. Based on our experience, it is not so much the sensitivity or specificity that influences the value of the estimate as the robustness of these rates over time. The robustness of the estimates is driven by the correlation of data with the real (but unknown) number of cases in the population. However, the Reviewer proposes a completely different approach to the one we followed in this paper, which is based on real data. We still think that the findings under real conditions of estimation of the reproduction numbers are valuable and interesting results.

Line 106: Where is the data from these data bases made available (to ensure reproducibility)?

Authors’ answer: We are not sure that we fully understand the question of the Reviewer, which concerns the part of the manuscript devoted to the ethical conditions of access to the data. Of course, access of raw data (individual data) of the 3 information systems (SI-DEP, OSCOUR and SIVIC) is only granted to authorised persons, mainly health agency staff. 

However, what we use to make our estimates is the daily count of the number of cases recorded by these systems. It is these statistics that we provide and that are needed to estimate the number of reproductions (available in the data directory on the GitHub repository, .rds extension that you can read with R or simply using the code provided). All materials are given and all the work in the manuscript is reproducible. You can also use your own data: just new observed cases at successive times. 

Line 112-113: Please provide version numbers and settings used (seeds for pseudo-random number generators etc.)

Authors’ answer: The version number (4.1.1) has been added on the revised manuscript. No specific settings are used. No seeds for pseudo-random number generators were necessary as we applied the parametric method for estimates. The parameters (sliding windows and serial interval) were detailed in the manuscript and used in R code provided on the GitHub repository. 

Line 114: A citation is needed for this as well as an explanation of which metrics this statement is made according to

Authors’ answer: As stated by the Reviewer, we cited the paper of Gostic et al. (Gostic et al., 2020) and Nash et al. (Nash et al., 2022) in the revised manuscript.

Line 116-117: Why not use a weekly aggregated time series in place of this? If “near-real-time” estimation is required, this would presumably be alright (and remove the weekly oscillations in case counts)

Authors’ answer: Technically, it seems possible to calculate an estimate with weekly aggregated data if the serial interval is provided over the same time unit. However, the Cori method (like all the other alternative methods to our knowledge) is configured to process daily data and, as discussed by Nash et al. (Nash et al., 2022), the use of weekly aggregated data may have an impact on the quality of the estimates. In particular, the choice of the sliding window parameter in Cori’s method (smoothing parameter) would be tricky and based on assumptions that are not very credible (a constant value of Rt over more than one week). To our knowledge, no publication presents such estimates, and estimates based on weekly aggregate data remain an open question. 

Line 120: Provide formulae

Authors’ answer: The authors do not know exactly which formula the Reviewer is referring to. If it is the Gamma distribution of the serial interval, it is a standard statistical distribution. 

Or if the Reviewer is referring to the formulation of the estimator of Rt, it is best to refer to the original papers by Cori (Cori et al., 2013) or Thompson (Thompson et al., 2019) cited in the manuscript. 

Nonetheless, we have added a supporting information file that contains some formal details on the Cori’s method, which can be welcome for the Reviewer and other interested readers. 

Line 121: Authors should explain why they chose a 95% level

Authors’ answer: The 95% threshold is the usual threshold for providing a confidence interval (or credibility interval, since we are in the context of a Bayesian calculation) for an estimate. All published estimates are given with this standard threshold, so there was no justification for using another threshold.

Line 130-139: This is not a result but rather a description of the data inputs. Accordingly, it should be moved to the methods section.

Authors’ answer: As the interpretation of the variation in the Rt estimates is interesting along with this description, the authors believe that it is relevant to keep this part in the results. Furthermore, we can subjectively consider that the statistical description of the data is a result and not formally an element of the method, the way in which the data is collected. 

Line 135-136: This is more so true for the OSCOUR and SI-VIC data sets which could be mentioned

Authors’ answer: As suggested by the reviewer, we modified the manuscript accordingly.

Line 137: The format of the lockdowns should be explained as this is country-specific and the reader may not know what restrictions were in place in France.

Authors’ answer: As suggested by the Reviewer, we added some details on the lockdown periods in France in the revised manuscript. 

The lockdown from October 29 to December 14, 2020 imposed confinement measures like movement restrictions, closure of non-essential shops but schools remained open. The lockdown between April 3rd and May 2, 2021 was less restrictive for movements but imposed closure of schools.

Line 137-139: As this is outside of the study period, it should be removed

Authors’ answer: The sentence has been deleted in the revised manuscript. 

Line 146-147: See previous comment

Authors’ answer: The sentence has been deleted accordingly.

Line 153-154: But the intervals seem to be non-existent the rest of the time. Is that plausible?

Authors’ answer: The width of the interval depends on the number of daily cases. The greater the number of daily cases, the smaller the variance of the posterior distribution of Rt. Again, for more details on the statistical framework of the method, we refer the reviewer to the supplementary material by Cori (Cori et al., 2013) or the paper of Thompson (Thompson et al., 2019) and to the supporting information file. For SI-DEP, the average number of positive cases per day was 40,000 over the study period, up to 500,000 positive cases in January 2022, which necessarily leads to extremely narrow credibility interval.

Line 162: When this has a temporal dimension, why not show it?

Authors’ answer: We are afraid that we do not fully understand the meaning of the Reviewer's comment. In this section, we show only the absolute differences observed between the estimates from each data source at each point in time.

Line 169-171: Provide better axis labels

Authors’ answer: We propose new axis label in the revised manuscript for Figure 3. 

Line 172: Which type of correlation is used. Provide the formulae

Authors’ answer: Implicitly, without precision, it is a Pearson correlation. We have specified this in the revised manuscript.

Line 182: But how confident can we be that the signal is correct?

Authors’ answer: There is no reason to suspect that the Rt estimate based on the results of the biological tests, either in relation to estimates from other data sources (taking into account the time lag) or in relation to the actual situation of the epidemic, has led to an erroneous signal about the spread of the epidemic. 

Line 182-191: This intuition could be laid out in the beginning

Authors’ answer: Indeed, the comparison of the Rt estimates obtained is consistent with the nature of the data, which once again reinforces confidence in the results obtained.

Line 197: When is “a few months at the start of the pandemic”? Please provide dates

Authors’ answer: We can assume that not all cases reported by ED’s were tested until the end of 2020, before antigen tests became widely available in France. We add this information in the 

revised manuscript as suggested by the Reviewer.

Line 216-218: But how did you quantify this?

Authors’ answer: Since May 2020, we have been producing the Rt as well as other epidemiological indicators used by experts and the French authorities to contain the spread of the virus within the population. The variations in Rt estimated have always been consistent with these other indicators and have been used to anticipate changes in incidence and adapt measures. 

Line 227-228: Please provide dates for the emergency for France since this is context-specific (depends on local conditions)

Authors’ answer: If this comment is linked to the date of appearance of the Omicron variant, we have added that this variant appeared in France at the end of November 2021 in the revised manuscript.

Line 228-230: Why not do a sensitivity analysis? Change point analysis? Interrupted time series analysis?

Authors’ answer: We have not shown the sensitivity analysis on the serial interval but we have described the consequences of the choice of the mean and standard deviation of the serial interval on the Rt estimates in the manuscript discussion. Indeed, the SI has an impact on the value of the Rt estimates but does not affect the temporal shape. In the first figure in the Supporting information file, we show this shape for the estimates using the SI-VIC data for 3 values of the mean for the gamma distribution: 3, 5.2 and 7 with a standard deviation of 4.7. As expected, Rt becomes higher as the mean of the serial interval increase

In addition, the higher the standard deviation of the serial interval, the lower the Rt estimates (second figure in the Supporting information file):

However, in practice, to monitor the spread of the epidemic in (near) real time, as indicated in the manuscript, we were more interested in the temporal variation in Rt than in its absolute values. The parameters of the serial interval distribution being of less importance in this case. In addition, in this article, as our objective was simply to compare estimates from the three data sources, all other things being equal, we have of course kept the same parameters for the three estimates. 

In response to the reviewer's suggestion, we propose to add a supporting information file to illustrate this particular point. 

With regard to the other suggestions made by the Reviewer (analysis of breakpoints), the authors did not understand well the purpose of these analyses. If this comment concerns the analysis of temporal variations in parameters of the serial interval, we unfortunately did not have the data to do this work. 

Line 235: What does “less responsive” mean?

Authors’ answer: Our goal is to detect change in Rt estimates, “responsive” meaning we are able to detect this change as early as possible. 

Line 242-243: But you noted in line 207-209 that there was a change in the ascertainment (which is super interesting) as your reasoning so that does not seem consistent with this assumption

Authors’ answer: In lines 207-209, we point out that during a period of about 3 months; this assumption was probably no longer true for the SI-DEP data (with explanation given in the manuscript). But, outside this period, the hypothesis can still be considered reasonable. As the Reviewer rightly pointed out above, this shows the importance of interpreting the Rt in the light of the other epidemiological indicators available.

Line 246-247: Would this be included in a Sante Publique France early warning system?

Authors’ answer: Indeed, nowadays Santé publique France has integrated the Rt estimated solely from Oscour into the Covid-19 surveillance system at national and infra-national level. And as mentioned above, Rt from syndromic surveillance data is currently being integrated into the surveillance of seasonal influenza epidemics and, in the near future, bronchiolitis. 

Line 261: What about waste water monitoring? Flunet? Other non-invasive options for estimating population-level incidence and prevalence exist

Authors’ answer: Already used in other countries to detect the circulation of viruses, a system to detect early pathogens in wastewater in France is planned since 2021 but not yet fully operational 

at this day: the SUM'eau project. This microbiological monitoring system for wastewater is currently in a transitional phase and involves 12 treatment plants in major cities which are monitored. The microbiological analysis results are sent to Santé publique France to produce one of the indicators for the Covid-19 monitoring system. A network of laboratory samples such as Flunet, which feeds a database, could also be a solution, but is not yet envisaged.

Provide an author CReDIT (contributor roles taxonomy) statement

Authors’ answer: Contributor roles were detailed in the revised manuscript.

Conceptualization: Christophe Bonaldi, Anne Fouillet, Juliette Paireau

Data Curation: Anne Fouillet, Cécile Sommen

Software: Christophe Bonaldi, Juliette Paireau, Anne Fouillet 

Supervision: Juliette Paireau, Daniel Lévi-Bruhl

Validation: Christophe Bonaldi, Anne Fouillet, Cécile Sommen, Daniel Lévi-Bruhl, Juliette Paireau

Writing – Original Draft Preparation: Christophe Bonaldi

Writing – Review & Editing: Christophe Bonaldi, Anne Fouillet, Cécile Sommen, Daniel Lévi-Bruhl, Juliette Paireau

Formatting:

The authors should ensure consistency of $R_t$ throughout (R$t$ and R) seen as well as the naming (“instantaneous reproduction number” also used in place of “effective reproduction number”). Additionally, the use of R in the introduction is inconsistent with the abstract

Line 60: I would suggest the authors use the name Sante Publique France throughout (since it is introduced)

Line 77: I would suggest using the term “COVID-19 certificate” rather than “health pass” as the former is more known (being the term used at EU-level)

Line 81-82: It should be “EDs” since multiple departments are being referenced

Line 84: I would suggest authors used “triage” in place of “coding” to avoid confusion with the concept of coding as it pertains to software code

Line 90: I would suggest “health emergency” in place of “unusual health situation”

Line 141: This should be “RT-PCR” to be consistent with when the abbreviation was introduced

Line 149: R_t falling below 1 is “desired” rather than “expected”

Line 156: Should be “Fig 2”

Line 272: Should be “Imperial College”

In abstract should be date accessed rather than date cited

Line 299 and 305: What is “80-“ is there a superfluous hyphen?

Check colours are differentiable when printing in black and white

In figure 3 could include a dashed line at zero since this is the value of interest for this graph (though perhaps a table would be better)

Authors’ answer: All formatting imperfections have been corrected in the revised manuscript in accordance with the reviewer's comments.

Reviewer #2: This paper applies a method developed by Cori et al (2013) and implemented in the statistical software R to calculate the effective reproduction number Rt during the COVID pandemic in France (May 2020-March 2022) on a daily basis. This method requires some general information on the distribution of the serial interval for the COVID-19, as well as daily incidence data of COVID-19. For the latter, three distinct data sets were used, respectively the daily number of 1) confirmed cases (among those tested), 2) emergency department visits (with suspected COVID), and 3) hospital admissions (due to COVID). Although such data are necessarily incomplete, the Rt obtained were remarkably similar using any of the three data sets, illustrating the fact that a real-time surveillance can be carried out with imperfect data, a useful result for public health.

The paper is very well written and clearly publishable. A minor revision would make it even better. I have the following remarks:

Authors’ answer: We thank the Reviewer for his positive evaluation of our paper and his remarks that helped us to greatly improve our manuscript. We hope that the revised version adequately addresses his main concerns.

1) Statistical method: If feasible, a little more explanation about the method of Cori et al (possibly citing the related paper by Wallinga and Teunis, 2004, American Journal of Epidemiology, 160, 509-516) would be welcome for an interested reader.

Authors’ answer: We agree with the Reviewer that our reference list should be completed., Together with the publication by Wallinga and Teunis as suggested by the Reviewer, we have introduced the paper by Thompson et al. (Epidemics, 2019) describing the latest version of EpiEstim. We have also added papers by Gostic et al (published in Plos Computational Biology in 2020) and Nash et al (PLOS Digit Heal. 2022) who have investigated the performance of different methods, which may also be of interest to readers. In addition, we have detailed the Cori’s method (parametric version used in our study) in an additional information file.

2) Discussion: It is written that Rt becomes higher as the mean and standard deviation of the serial interval increase. While this is true for the mean (at least when Rt is greater than 1), Rt would actually decrease with a higher standard deviation, being e.g. overestimated in the extreme case where the standard deviation is wrongly set to 0 (see e.g. Wallinga & Lipsitsch, 2007, Proc. of the Royal Society B, 274, 599-604). This should be corrected.

Authors’ answer: We thank a lot the reviewer to notice that the sentence written in our manuscript was indeed incorrect. Consequently, we modified in the revised version. In addition – and as part of the request for a sensitivity analysis from a first reviewer - this point is detailed in the supporting information file.

3) As acknowledged by Cori et al (2013), the serial interval (time between onset of symptoms in the infector and the infected) is used as a proxy of the generation interval (time between infections of the infector and the infected) which should be ideally used in their method. As a direct consequence of my first comment, this should result in an underestimation of Rt since the standard deviation of the generation interval (which should be used) is generally smaller than that of the serial interval (which has been used, see also Ganyani et al, 2020, Euro Surveillance, 25, 2000257). This should be mentioned in the Discussion.

Authors’ answer: The reviewer raised a very interesting point and we agree that the impact of using the serial interval instead of the generation interval needs to be discussed. Therefore, we have completed our discussion to include this issue in our revised manuscript. We are very grateful to the Reviewer for the suggested article and we have also cited the article by Britton & Scalia Tomba, 2019 (Journal of the Royal Society Interface, 16 ) which provides welcome material for a reader interested in the biases generated by replacing generation times with serial intervals. 

4) In the Discussion, it is briefly mentioned that serial intervals could be shorter with the new Omicron variant. But a shorter serial interval would lead to a smaller Rt, although the Omicron variant was supposed to be much more contagious than the original one, reaching an average Rt of 3.4 (Liu and Rocklov, J. Travel Medicine, 2022, 29, 1-4). How do the authors explain that the peaks in their estimation of the reproduction number did not get that high when new (more contagious) variants appeared? This would deserve one (possibly short) paragraph in the Discussion.

Authors’ answer: The new Omicron variant appeared in France at the end of November 2021 and rapidly replaced the Delta variant, which was the majority variant. Although the Omicron variant is probably more contagious, Rt estimates in France never reach the value reported by Liu et al. It is of course difficult to determine the reasons for this observation with any certainty. However, it is reasonable to assume that the impact of vaccination (over 75% of the French population had full vaccination coverage by December 2021) limited the spread of the virus. The estimated Rt was therefore much lower than the peak observed in July 2021, when the delta variant was in the majority and vaccination coverage was lower. In addition, there were 2 major waves: the first starting in November, but probably still linked to the delta variant, and a rebound in mid-December linked to the end-of-year festivities and probably also to the spread of the Omicron variant. Here again, immunity induced by the vaccine or contamination probably helped to limit the spread of this variant. Finally, it should also be noted that preventive measures to limit the spread of the virus (masks on public transport, widespread teleworking, reception of a limited number of customers depending on the surface area in shops or subject to vaccination certificate, etc.) were still in force at that time in France (until 12 February 2021 for the most part). This may also have helped to limit the spread of the variant. 

As suggested by the Reviewer, we integrated this element in the discussion of the revised manuscript.

5) Results: “As expected, Rt fell below 1 after the implementation of nationwide lockdown”. It is in fact not obvious to see the effects of the lockdown on such a graphic. These were not the only occasions where Rt fell below 1. Note also that the reproduction number started to rise again before the end of the first lockdown, while it continued to fall well after the second lockdown. One or two sentences recognizing this could be an option (public health is a challenging science!).

Authors’ answer: We agree with the Reviewer that the impact of lockdowns on the Rt trend is not so easy to read! Lockdown periods have often been determined on the basis of the burden on the hospital system and when incidence levels were high in the population. In addition, unlike the first lockdown (from March, 17 to May 3, 2020), these periods were marked by a strengthening of measures (on movement or schools) that already existed in less strict forms outside these periods. During the second lockdown, schools remained open, which may explain the rebound before the end of the lockdown, when children were a major vector for the spread of COVID-19. This was not the case for the third lockdown, when schools were closed. We propose adding the following paragraph to our discussion of the revised manuscript:

“Over the entire period studied, estimates of the number of reproductions have fluctuated constantly. The largest waves are correlated with the emergence of new variants such as the Alpha variant at the end of 2020, the Delta variant in the summer of 2021 and the Omicron variant at the beginning of 2022. Contextual events such as the Christmas and New Year period could explain the peak observed in 2021. We also noticed that during the second lockdown, the number of reproductions had surprisingly rebounded after reaching a minimum value, returning to a value close to 1 when the lockdown ended. The second national lockdown was less strict than the first one (from March 17 to May 3, 2020), but all schools remained open. This could suggest that children and adolescents - who present a low risk of infection or if they are infected it is likely to be mild - may have been transmitters of the virus into the household during this period (Cox, 2023). It should be noted that we did not observe this phenomenon during the third lockdown, which was marked by a reversed strategy with the early closure of schools before the Easter school holidays.”

6) Caption Figure 2: Write Fig.2 instead of Fig.1. Would you also like to add the relevant dates here (May 2020-March 2022)?

Authors’answer: The relevant dates have been added in the revised manuscript. 

We are very grateful to the second Reviewer for his very interesting comments, which helped us to improve significantly the discussion of our article. We hope that our responses to the comments and the revision of the manuscript will be satisfactory.

References:

Britton, T., & Scalia Tomba, G. (2019). Estimation in emerging epidemics: biases and remedies. Journal of the Royal Society, Interface, 16(150), 20180670. https://doi.org/10.1098/rsif.2018.0670

Cori, A., Ferguson, N. M., Fraser, C., & Cauchemez, S. (2013). A new framework and software to estimate time-varying reproduction numbers during epidemics. American Journal of Epidemiology, 178(9), 1505–1512. https://doi.org/10.1093/aje/kwt133

Cox, D. (2023). What do we know about covid-19 and children? BMJ (Clinical Research Ed.), 380, 21. https://doi.org/10.1136/bmj.p21

Gostic, K. M., McGough, L., Baskerville, E. B., Abbott, S., Joshi, K., Tedijanto, C., Kahn, R., Niehus, R., Hay, J. A., De Salazar, P. M., Hellewell, J., Meakin, S., Munday, J. D., Bosse, N. I., Sherrat, K., Thompson, R. N., White, L. F., Huisman, J. S., Scire, J., … Cobey, S. (2020). Practical considerations for measuring the effective reproductive number, Rt. PLoS Computational Biology, 16(12), e1008409. https://doi.org/10.1371/journal.pcbi.1008409

Ma, Q., Liu, J., Liu, Q., Kang, L., Liu, R., Jing, W., Wu, Y., & Liu, M. (2021). Global Percentage of Asymptomatic SARS-CoV-2 Infections Among the Tested Population and Individuals With Confirmed COVID-19 Diagnosis: A Systematic Review and Meta-analysis. JAMA Network Open, 4(12), e2137257–e2137257. https://doi.org/10.1001/JAMANETWORKOPEN.2021.37257

Nash, R. K., Nouvellet, P., & Cori, A. (2022). Real-time estimation of the epidemic reproduction number: Scoping review of the applications and challenges. PLOS Digital Health, 1(6), e0000052. https://doi.org/10.1371/journal.pdig.0000052

Pullano, G., Di Domenico, L., Sabbatini, C. E., Valdano, E., Turbelin, C., Debin, M., Guerrisi, C., Kengne-Kuetche, C., Souty, C., Hanslik, T., Blanchon, T., Boëlle, P. Y., Figoni, J., Vaux, S., Campèse, C., Bernard-Stoecklin, S., & Colizza, V. (2020). Underdetection of cases of COVID-19 in France threatens epidemic control. Nature 2020 590:7844, 590(7844), 134–139. https://doi.org/10.1038/s41586-020-03095-6

Thompson, R. N. N., Stockwin, J. E. E., van Gaalen, R. D. D., Polonsky, J. A. A., Kamvar, Z. N. N., Demarsh, P. A. A., Dahlqwist, E., Li, S., Miguel, E., Jombart, T., Lessler, J., Cauchemez, S., & Cori, A. (2019). Improved inference of time-varying reproduction numbers during infectious disease outbreaks. Epidemics, 29, 100356. https://doi.org/10.1016/j.epidem.2019.100356

---

## [Decision Letter · Decision Letter 1]

17 Oct 2023

Monitoring the reproductive number of COVID-19 in France: Comparative estimates from three datasets

PONE-D-22-25321R1

Dear Dr. Bonaldi,

We’re pleased to inform you that your manuscript has been judged scientifically suitable for publication and will be formally accepted for publication once it meets all outstanding technical requirements.

Kind regards,

Kaspar Staub

Academic Editor

PLOS ONE

Additional Editor Comments (optional):

Reviewers' comments:

Reviewer's Responses to Questions

**Comments to the Author**

1. If the authors have adequately addressed your comments raised in a previous round of review and you feel that this manuscript is now acceptable for publication, you may indicate that here to bypass the “Comments to the Author” section, enter your conflict of interest statement in the “Confidential to Editor” section, and submit your "Accept" recommendation.

Reviewer #2: All comments have been addressed

2. Is the manuscript technically sound, and do the data support the conclusions?

Reviewer #2: Yes

3. Has the statistical analysis been performed appropriately and rigorously? 

Reviewer #2: Yes

4. Have the authors made all data underlying the findings in their manuscript fully available?

Reviewer #2: Yes

5. Is the manuscript presented in an intelligible fashion and written in standard English?

Reviewer #2: Yes

6. Review Comments to the Author

Reviewer #2: The authors did a good job in revising this manuscript. I am happy with the revised maanuscript and have no further comments.

7. PLOS authors have the option to publish the peer review history of their article (what does this mean?). If published, this will include your full peer review and any attached files.

Reviewer #2: No

---

## [Editor Report · Acceptance letter]

23 Oct 2023

PONE-D-22-25321R1 

Monitoring the reproductive number of COVID-19 in France: Comparative estimates from three datasets 

Dear Dr. Bonaldi:

I'm pleased to inform you that your manuscript has been deemed suitable for publication in PLOS ONE. Congratulations! Your manuscript is now with our production department. 

Kind regards, 

on behalf of

PD Dr. Kaspar Staub 

Academic Editor

PLOS ONE